# Assessing the Need for Multiplex and Multifunctional Tick-Borne Disease Test in Routine Clinical Laboratory Samples from Lyme Disease and Febrile Patients with a History of a Tick Bite

**DOI:** 10.3390/tropicalmed6010038

**Published:** 2021-03-17

**Authors:** Kunal Garg, T. Sakari Jokiranta, Sanna Filén, Leona Gilbert

**Affiliations:** 1Tezted Ltd., Mattilaniemi 6-8, 40100 Jyväskylä, Finland; 2United Medix Laboratories, Kivihaantie 7, 00310 Helsinki, Finland; sakari.jokiranta@tammerbiolab.fi (T.S.J.); sanna.filen@gmail.com (S.F.)

**Keywords:** Lyme disease, tick-borne disease, zoonoses, spirochetes, polymicrobial, summer flu, misdiagnosis, persister, Borrelia, Lyme diagnostic

## Abstract

Human polymicrobial infections in tick-borne disease (TBD) patients is an emerging public health theme. However, the requirement for holistic TBD tests in routine clinical laboratories is ambiguous. TICKPLEX^®^ PLUS is a holistic TBD test utilized herein to assess the need for multiplex and multifunctional diagnostic tools in a routine clinical laboratory. The study involved 150 specimens categorized into Lyme disease (LD)-positive (*n* = 48), LD-negative (*n* = 30), and febrile patients from whom borrelia serology was requested (*n* = 72, later “febrile patients”) based on reference test results from United Medix, Finland. Reference tests from DiaSorin, Immunetics, and Mikrogen Diagnostik followed the two-tier LD testing system. A comparison between the reference tests and TICKPLEX^®^ PLUS produced 86%, 88%, and 87% positive, negative, and overall agreement, respectively. Additionally, up to 15% of LD and 11% of febrile patients responded to TBD related coinfections and opportunistic microbes. The results demonstrated that one (TICKPLEX^®^ PLUS) test can aid in a LD diagnosis instead of four tests. Moreover, TBD is not limited to just LD, as the specimens produced immune responses to several TBD microbes. Lastly, the study indicated that the screening of febrile patients for TBDs could be a missed opportunity at reducing unreported patient cases.

## 1. Introduction

Lyme disease (LD) is a tick-borne disease (TBD) caused by bacteria from the *Borrelia burgdorferi sensu lato* group that can cause arthritic, dermatitis, or neurological manifestations [1,2,3,4]. Other common TBDs also include Babesiosis, Ehrlichiosis, Anaplasmosis, Encephalitis, and more [5,6,7,8]. Currently, TBDs are present in over 80 countries and may affect 35% of the world’s population by 2050 [9]. In the meantime, the number of ticks that carry pathogens and can cause TBDs are ever-increasing [10,11,12,13]. Over the years, the reported TBD cases have spiked in various countries around the world [14,15,16]. Healthcare authorities like the Centers for Disease Control and Prevention (CDC) in the USA recognize that the real frequency of TBD cases in humans is much higher than the reported cases [17]. In 2018, the European Commission made headway by adding Lyme Neuroborreliosis to the list of diseases under the European Union’s epidemiological surveillance [18]. Additionally, the European Parliament resolution recognized that the current TBD diagnostic tools are inaccurate, as they test for only one microbe at a time [19].

Globally, the CDC two-tier testing algorithm for LD stands undisputed by regulatory and healthcare authorities [20]. The literature is rife with evidence concerning the effectiveness of the CDC two-tier system for diagnosing LD [1,21,22]. The CDC recently revised its LD testing algorithm by endorsing the use of two enzyme-linked immunosorbent assays (ELISAs) in both tiers [23]. However, the testing recommendations for other TBDs in LD patients is not clear, despite the growing evidence of coinfections in such patients [24]. An estimated 85% of LD patients can produce an immune response to TBD-related coinfections or opportunistic microbes [25]. Yet, 83% of all commercial TBD tests—for example, in the USA—are solely prescribed for LD [26]. The most putative diagnostic test manufacturers have popularized the use of a single test for a single disease following the Germ Theory [21,22,27]. As a result, the role, relevance, and requirements for a multiplex and multifunctional tool in the diagnosis of a complex disease like TBD are unclear for routine use in clinical laboratories.

Internationally, the research community has confirmed the likelihood of immune dysfunction in LD patients due to pathogenesis by Borrelia [28,29,30,31,32,33]. A TBD patient may experience an increase in disease severity, as Borrelia can sabotage, undermine, or trick the host immune system by evasion [32,33,34]. For example, Borrelia can repress the antigen-induced proliferation of lymphocyte cells or anti-Borrelia antibody response in immunocompromised patients [28,33]. Additionally, Borrelia can meddle with the kinetics and quality of B-cell and T-cell responses [34,35]. Hence, LD patients can present seronegative, delayed, or persistent antibody responses to Borrelia, indicating the complex nature of TBDs and a possible reason for misdiagnosed or undiagnosed cases [35,36,37,38]. Additionally, the regular discovery of novel and emerging TBD pathogens such as *Rickettsia monacensis,* Powassan virus, Omsk hemorrhagic virus, and others further complicates treatment for TBD patients without a holistic diagnostic tool [39].

A holistic diagnostic test may also help realize the need to institute a differential diagnosis in TBD testing recommendations. Patients with common symptoms like fever, headache, cough, and chills in the absence of laboratory evidence for LD could be misdiagnosed or remain undiagnosed for other conditions [40,41,42]. The prevalence of well-known TBD-related coinfections and opportunistic microbes are evident in individuals suffering from myalgia, fatigue, arthritis, and more [43]. For example, infection with *Bartonella* species can cause patients to complain about myalgia and fatigue [44]. Similarly, patients with fibromyalgia and chronic fatigue syndrome demonstrate an immune response to *Mycoplasma pneumoniae* or *Mycoplasma fermentans [45,46]*. While TBD is complicated to diagnose according to the literature mentioned above, will the use of comprehensive diagnostic tests prove practical to help reduce unrecognized patient cases? The goal of this study was to assess the need for a multiplex and multifunctional TBD immunoassay in routine clinical laboratory samples from Lyme disease and febrile patients a with (suspected) history of a tick bite.

## 2. Materials and Methods

### 2.1. Index Test and Interpretation

TICKPLEX^®^ PLUS (herein, TICKPLEX^®^) is an ELISA index test used in this study that is a CE-IVD registered product (i.e., European In-Vitro Diagnostic Devices Directive (98/79/EC) compliant) manufactured in an ISO 13485:2016 accredited facility at Tezted Ltd, Jyväskylä, Finland. TICKPLEX^®^ can measure the immunoglobulin M (IgM) and immunoglobulin G (IgG) immune responses in human serum samples against *Borrelia burgdorferi sensu lato* species in spirochete and persistent forms, coinfections, and opportunistic microbes. Mainly, TICKPLEX^®^ includes *Borrelia burgdorferi sensu stricto*, Borrelia afzelii, and *Borrelia garinii* in spirochete and persistent form, *Babesia microti*, *Bartonella henselae*, *Ehrlichia chaffeensis*, *Rickettsia akari*, Coxsackievirus, Epstein–Barr virus, Human parvovirus B19, *Mycoplasma fermentans*, and Mycoplasma pneumoniae [25]. The clinical relevance for all TICKPLEX^®^ microbes in TBD patients has been previously demonstrated [25]. The 150 human serums were tested blindly with the index test at Tezted Ltd. Normalized optical density values at 450 nm lower than 0.90, between 0.91 to 0.99, and higher than 1.00 were negative, borderline, and positive immune responses for all microbes, respectively.

### 2.2. Ethics Statement

United Medix Laboratories (Finland) provided anonymized and leftover human sera samples for research purposes. Sera sample included reference test results for LD, age, and gender for all patients. Following the General Data Protection Regulation (GDPR) [47], researchers at Tezted Ltd. did not have access to any private information (i.e., name, profession, or ethnicity) from the specimens that could be linked back to the patients. Hence, following the Declaration of Helsinki embodied in Common Rule set forth by the Code of Federal Regulations, USA, informed consent was not collected, as the present study was not considered as human subject research [48,49]. In Finland, the medical research act (488/1999) and the law on the medical usage of human organs, tissues, and cells (2.2.2001/101; section 20 (30.11.2012/689)) supports the use of leftover and deidentified human serum samples with consent from the collection unit [50,51]. United Medix Laboratories (Finland) was the collection unit for this study that contributed the deidentified human serum specimens according to their International Organization for Standardization (ISO) 15189 section 5.9.1. quality management system [52].

### 2.3. Reference Tests and Interpretation

Healthcare providers in Finland follow the CDC two-tier guidelines for LD diagnosis. Thus, Diasorin LIAISON^®^ Borrelia chemiluminescence immunoassay (CLIA), Immunetics^®^ C6 Lyme ELISA^TM^ (C6 ELISA), and Mikrogen Diagnostik *recom*Bead Borrelia IgG 2.0 (IgG Blot) were used to confirm LD in human specimens. The CLIA test separately measures human IgM and IgG immune responses to *Borrelia burgdorferi sensu lato.* In contrast, the C6 ELISA measures human IgM and IgG combined immune reactions to the C6 synthetic peptide derived from the VlsE protein conserved in *Borrelia burgdorferi sensu stricto* or *Borrelia afzelii* and *Borrelia garinii*. For LD confirmation purposes, IgG Blot measured the human IgG immune response against *Borrelia burgdorferi sensu stricto*, *B. garinii*, *B. afzelii*, *B. bavariensis*, and *B. spielmanii*.

For the CLIA IgM test, arbitrary units per milliliter (AU/ml) less than 18, between 18 to 22, and more than 22 were considered negative, borderline, and positive immune responses, respectively. Similarly, for the CLIA IgG test, AU/ml less than 10, between 10 to 15, and more than 15 were considered negative, borderline, and positive immune responses, respectively. Like AU/ml, the C6 ELISA test utilized the Lyme Index (LI) with a normalized optical density value at 450 nm and a reference wavelength at 650 nm. As a result, LI less than 0.9, between 0.91 to 1.09, and more than 1.10 were considered negative, borderline, and positive immune responses, respectively. In the case of the IgG Blot test, normalized fluorescence intensities below 0.67, between 0.67 to 1.00, and above 1.00 were considered negative, borderline, and positive immune responses, respectively.

### 2.4. Patient Categorization

According to the CDC two-tier algorithm [24] for LD diagnosis and related test interpretation criteria, as mentioned above, the 150 human serum samples were organized in three different categories. LD-positive category (*n* = 48) included specimens with positive IgM or IgG immune responses to one (*n* = 7), two (*n* = 17), three (*n* = 9), or all four (*n* = 15) diagnostic tests. Category two included LD-negative (*n* = 30) serum samples with a negative immune response to all four tests (*n* = 15) and a positive immune response limited to the CLIA IgM or IgG test (*n* = 15). The last category included serum samples from patients with fever and a known or suspected history of a tick bite, i.e., from whom borrelia serology was requested (later, the febrile patient group) (*n* = 72). For the febrile patient group, the test results from the C6 ELISA and IgG Blot tests were not available. 

### 2.5. Index Test and Interpretation

TICKPLEX^®^ PLUS (herein, TICKPLEX^®^) is an ELISA index test used in this study that is a CE-IVD registered product manufactured in an ISO 13485:2016 accredited facility at Tezted Ltd. TICKPLEX^®^ can measure IgM and IgG immune responses in human serum samples against *Borrelia burgdorferi sensu lato* species in spirochete and persistent forms, coinfections, and opportunistic microbes. Mainly, TICKPLEX^®^ includes *Borrelia burgdorferi sensu stricto, Borrelia afzelii,* and *Borrelia garinii* in spirochete and persistent form, *Babesia microti*, *Bartonella henselae*, *Ehrlichia chaffeensis*, *Rickettsia akari*, Coxsackievirus, Epstein–Barr virus, Human parvovirus B19, *Mycoplasma fermentans*, and Mycoplasma pneumoniae [25]. The clinical relevance for all TICKPLEX^®^ microbes in TBD patients has been previously demonstrated [25]. The 150 human serums were tested blindly with the index test at Tezted Ltd. Normalized optical density values at 450 nm lower than 0.90, between 0.91 to 0.99, and higher than 1.00 were negative, borderline, and positive immune responses for all microbes, respectively.

### 2.6. Statistical Analysis

For quality control purposes, an inter-plate and inter-operator precision analysis was conducted by assessing the coefficient of variance [53] (CV %) on the optical density values for IgM/IgG plate controls and all microbial antigens on TICKPLEX^®^. To assess the CV % for index test microbial antigens, the negative serum control (TEZ1) in the kit was repeatedly performed in each plate by each operator. Equations (1)–(3) were utilized to calculate the proportion of positive (PA), negative (NA), and overall (OA) agreement, respectively, among the reference tests and between the reference tests versus (vs.) index test [54]. The PA, NA, and OA agreements among the reference tests and between reference tests with the index test were combined for the IgM and IgG immune responses. In Equations (1)–(3), the letters a, b, c, and d stand for true positives, false positives, false negatives, and true negatives, respectively. Further, the reliability for each PA and NA comparison was evaluated by calculating Cohen’s kappa (*k*) with a 95% confidence interval [54,55].
(1)$$\mathrm{PA}=\;\frac{2a}{2a+b+c}$$
(2)$$\mathrm{NA}=\;\frac{2d}{2d+b+c}$$
(3)$$\mathrm{OA}=\;\frac{a+d}{a+b+c+d}$$

Cohen’s *k* ranges from −1 to +1, wherein *k* values ≤ 0 indicates no agreement, 0.01–0.20 as none to a slight agreement, 0.21–0.40 as fair agreement, 0.41–0.60 as moderate agreement, 0.61–0.80 as substantial agreement, and 0.81–1.00 as almost perfect agreement [55]. Proportionate positive and negative agreements, along with Cohen’s *k*, were calculated using the EPITOOLS diagnostic test evaluation and comparison calculator. The inter-rater reliability and proportional agreement analysis between various tests were carried out using just LD-positive and -negative patient groups. Further, Fisher’s exact test was used to assess the statistical differences in IgM or IgG immune responses between the LD (positive and negative) and febrile patient groups. The two-tailed *p*-values for the Fisher’s exact test were calculated using GraphPad (https://www.graphpad.com/quickcalcs/contingency1/ (accessed on 28 May 2019). Fisher’s exact test results with *p*-values < 0.05 were considered statistically associated or dependent [56].

## 3. Results

The United Medix Laboratories in Finland collected specimens from LD-positive (*n* = 48) and LD-negative (*n* = 30) patients and from febrile patients from whom borrelia serology was requested (*n* = 72). The samples were collected amid routine clinical diagnostic services (convenience sampling) in the summer of 2018, beginning from late-May to mid-September. On average, patients were 42, 39, and 36 years old in the LD-positive, LD-negative, and febrile groups, respectively. The LD-positive patient group included 27 male and 21 female human serum samples. Likewise, the LD-negative group included specimens from 15 male and 15 female patients. Lastly, specimens from the febrile patients consisted of 31 male and 41 female human specimens. Overall, the average age for 73 male and 77 female serum samples was 39 years. Further, the inter-plate and inter-operator CV % for IgM and IgG on the index test were 6.280% and 4.692%, respectively. Additionally, the CV % for the internal negative control (TEZ1) was observed to be ≤15% for all microbial antigens on the index test.

Figure 1 illustrates the PA, NA, OA, and Cohen’s *k* among the reference tests and between the reference tests with the index test. The PA for the individual reference or index test ranged between 53% for IgG Blot vs. TICKPLEX^®^ to 72% for CLIA IgM/IgG vs. C6 ELISA. Similarly, the lowest NA was observed for CLIA IgM/IgG vs. TICKPLEX^®^ (49%) and the highest between IgG Blot vs. TICLPLEX (76%). The OA ranged from 55% for CLIA IgM/IgG vs. TICKPLEX^®^ to 73% between C6 ELISA vs. IgG Blot. Except for a moderate Cohen’s *k* agreement between C6 ELISA vs. IgG Blot (*k* = 0.45), all the other individual test combinations displayed fair Cohen’s *k* agreements (*k* = 0.12 to 0.31). Among the different test comparisons individually, the average PA, NA, and OA were 63.5%, 62.33%, and 64%, respectively. As mentioned earlier, four Lyme disease tests (i.e., reference tests) were used to confirm Borrelia infection according to the CDC two-tier criteria. A substantial Cohen’s *k* agreement was observed between the commercial two-tiered tests vs. TICKPLEX^®^ (*k* = 0.74). The PA, NA, and OA for comparisons between all reference tests and TICKPLEX^®^ were 86%, 88%, and 87%, respectively (Figure 1).

In addition to Lyme disease, the LD-positive, LD-negative, and febrile patient groups were also tested against TBD related coinfections and opportunistic microbes using TICKPLEX^®^. Figure 2 is a cooccurrence heat map indicating the percentage of IgM or IgG immune responses by LD (positive and negative) and febrile patient groups to TICKPLEX^®^ antigens. Borrelia spirochete species and persistent forms witnessed the most significant percentage of IgM and IgG immune responses in both patient groups. Apart from Borrelia, an average 2% for IgM and 8% for IgG immune responses were noted by LD specimens to coinfections and opportunistic microbes related to TBD (herein other microbes). Likewise, on average, 4% for IgM and 6% for IgG immune responses were observed for febrile patient samples against other TBD-related microbes. Overall, a statistical association or dependence was observed between LD and the febrile patient group’s IgM and IgG responses to the Epstein–Barr virus and Borrelia spirochete species, respectively (Appendix A). No association in IgM or IgG immune responses with the remaining TICKPLEX^®^ antigens were noted between the LD and febrile patient groups (Appendix A).

The IgM and IgG immune responses by the LD and febrile patient groups to Borrelia alone, Borrelia and other microbes, and just other microbes in the index test were further analyzed (Figure 3). For IgM, 1%, 6%, and 0% LD patients responded to only Borrelia, Borrelia and other microbes, and only other microbes, respectively (Figure 3A). Similarly, 4%, 7%, and 4% of the febrile patient specimens produced an IgM response against only Borrelia, Borrelia and other microbes, and only other microbes, respectively (Figure 3A). In the case of IgG immune responses by LD patients, 28%, 15%, and 4% of the patients responded to only Borrelia, Borrelia and other microbes, and only other microbes, respectively (Figure 3B). Likewise, 14%, 11%, and 0% of the febrile patient specimens produced IgG response against only Borrelia, Borrelia and other microbes, and only other microbes, respectively (Figure 3B). A statistical association was observed between the LD and febrile patient groups’ IgG responses to only Borrelia (Figure 3).

Appendix A demonstrates the percentage of LD or febrile patient IgM and IgG immune responses to the number of other microbes along with Borrelia. The IgM or IgG immune responses to Borrelia and one other microbe was the most significant percentage of the reaction seen in both the LD and febrile patient groups. In the case of the LD patient group, 3% for IgM and 6% for IgG responded to Borrelia and one other microbe, respectively. Similarly, 4% and 3% of the febrile patients produced IgM and IgG responses to Borrelia and one other microbe, respectively. Not more than 1% of the LD or febrile patient specimens in IgM or IgG responded to Borrelia and two other microbes to seven other microbes. 

Remarkably, the second most significant percentage of IgM or IgG immune responses was seen in both the LD and febrile patient specimens for Borrelia and eight other microbes. Approximately 4% IgM or IgG immune responses were noted from the LD and febrile patient groups against Borrelia and eight other microbes. An IgM or IgG immune response to Borrelia and eight other microbes primarily responded to all ten TICKPLEX^®^ antigens. At random, a serum sample with IgM and IgG immune response to Borrelia and eight other microbes was selected and serially diluted on TICKPLEX^®^. As a result, a clear dose-dependent response was observed (Appendix A).

## 4. Discussion

To evaluate the use for a multiplex and multifunctional TBD immunoassay in a routine clinical laboratory, LD-positive (*n* = 48), LD-negative (*n* = 30), and febrile (*n* = 72) patient specimens were tested against TICKPLEX^®^ microbial antigens for their IgM and IgG immune responses. The clinical performance of TICKPLEX^®^ (index test) for testing LD was compared to four reference tests (CLIA IgM and IgG, C6 ELISA, and IgG Blot) used at the United Medix Laboratories in Finland following the CDC two-tier criteria. Individual comparisons among the reference tests and between the reference with the index tests resulted in an average PA, NA, and OA of 63.5%, 62.33%, and 64%, respectively (Figure 1). A substantial Cohen’s *k* agreement (*k* = 0.74) was mainly observed when the clinical outcome from all four reference tests was compared with the TICKPLEX^®^ results (Figure 1). A comparison between the commercial CDC two-tiered LD testing system with TICKPLEX^®^ produced 86% PA, 88% NA, and 87% OA (Figure 1).

Variations in the PA, NA, or OA among the LD diagnostic tests is a rule rather than the exception, because several in vitro diagnostic test manufacturers utilize different Borrelia proteins [21,22,27,57]. For example, the positivity rate for LD patients with an Erythema Migrans rash can range from 18% to 53% for whole-cell antigen LD tests vs. 31% to 50% for recombinant antigen LD tests [57]. Generally, diagnostic test sensitivities improve from the early to late LD stages [21,22,57]. In later LD stages like neuroborreliosis, the positivity rate can vary from 41% to 86% for whole-cell antigen LD tests and 49% to 81% for recombinant antigen LD tests [57]. Additionally, with regards to the CDC two-tier testing system, a PA among commercial LD tests can vary from 5% to 98.5%, and a NA can range from 28.6% to 100% [21]. Overall, at any given LD stage, the average accuracy for LD diagnostic tests is 62.3% [21,27]. Similar accuracy averages in this study were observed among the reference tests and between the reference and index tests (Figure 1).

While the accuracy averages for LD diagnostic tests between this study and the literature are comparable, the study findings herein also indicated that TICKPLEX^®^ is a suitable replacement for the CLIA IgM/IgG, C6 ELISA, and IgG Blot reference tests. A dramatic increase in correlations between the commercial CDC two-tiered LD tests and TICKPLEX^®^ is connected to a consistent PA (60% to 63%) with CLIA IgM/IgG and C6 ELISA plus a high NA (76%) with the IgG Blot test (Figure 1). The C6 ELISA demonstrates a similarly dramatic change in a PA and NA when compared with either an individual LD test or a CDC two-tiered testing system [21]. A previous comparison between the C6 ELISA and CLIA IgM/IgG tests yielded 70% PA and 99.1% NA [21]. However, the current study demonstrated 72% PA and only 54% NA between the C6 ELISA and CLIA IgM/IgG tests (Figure 1). Nevertheless, a 98.5% PA and 49% NA was evident between the C6 ELISA and CDC two-tiered tests, which included the Wampole Bb (IgG/IgM) ELISA test system, MarDx Lyme Disease (IgG and IgM), and Marblot strip test system [21]. As a result, the PA, NA, and OA of LD tests strongly depend on the type of reference test used for comparison [21,22,27,57].

The TICKPLEX^®^ results also indicated that 6% to 15% of the LD individuals responded to TBD-related coinfections and opportunistic microbes (Figure 2). Traditionally, a TBD-linked opportunistic infection in a LD patient could be the result of a vulnerable immune system due to a prolonged TBD infection [32,33,58]. Immune responses by LD patients to multiple other TBD microbes with or without Borrelia demonstrate that TBD is not limited to just LD in Finland (Figure 2 and Figure 3). In several other countries, like Germany, Sweden, the Netherlands, and more, 4% to 60% of LD patients can suffer from LD and TBD-related coinfections [59,60,61]. Multiple TBD-associated infections in LD patients primarily originate from ticks that can carry over 120 distinct bacterial and other microbial species [62]. In various regions of Finland, the cooccurrence percentage for multiple pathogens in ticks ranges from 1.02% to 28.3% [11,12,13,63,64]. In 2004, a Finnish LD patient suffered from fatal Babesiosis [65]. Therein, no research articles on PubMed elucidated the relevance of TBD-related coinfections or opportunistic microbes in Finland.

Furthermore, the IgG immune responses were statistically correlated between the LD and febrile patient groups (Appendix A and Figure 3). Moreover, 7% to 11% of the febrile patients reacted to other TBD-related microbes (Figure 2 and Figure 3). The current study demonstrated that individuals with fever and a putative history of a tick bite can respond to TBD microbes similar to LD patients (Figure 2 and Figure 3). A misdiagnosis of early LD as summer flu is an understudied topic in the field of TBDs [41]. Not all LD patients demonstrate an Erythema Migrans (EM) rash or produce detectable antibodies in the first two to four weeks. A misdiagnosis is probable for nearly 16% of LD cases that do not display an EM rash [66]. Additionally, 60% of early-stage LD individuals receive a negative LD diagnostic test result, as they do not develop a detectable level of antibodies and are therefore susceptible to misdiagnosis [67]. A TBD infection can cause nonspecific febrile illness wherein individuals may suffer from LD (11%), human granulocytic ehrlichiosis (13%), or coinfections (3%) [68].

Lastly, an IgM or IgG immune response to all TICKPLEX^®^ antigens by 4% of the LD and febrile patient groups is an unexpected finding in this study (Appendix A). The unspecific binding of human specimens to recombinant proteins or blocking agents on an ELISA test is a plausible interpretation [69,70]. However, all TICKPLEX^®^ antigens comprise of either whole-cell lysates or synthetic peptides and not recombinant proteins. Secondly, a sera sample from 4% of the LD and febrile patients at random was serially diluted to correlate the declining antibody concentration with optical density values. In the presence of an unspecific reaction, a serial dilution of sera specimen will not make any difference on the resulting optical density value. Appendix A indicates no unspecific binding on TICKPLEX^®^ for IgM and IgG. Immune evasion and host immune response suppression, modulation, or subversion by Borrelia in LD patients is a common finding [28,30,32,33,71,72,73]. For example, Borrelia can trick the host immune system into producing a strong yet inadequate response while it evades the lymph nodes [34]. We postulate that a universally positive IgM or IgG immune response in TBD patients could be the result of a B-cell-related immune dysfunction, such as unspecific B-cell activation [29,34,74].

A noticeable improvement to the current study would be to increase the overall sample size and improve the statistical confidence in the findings. In the future, the study design could also include a comparison between TICKPLEX^®^ non-Borrelia antigens and related reference tests in a routine lab clinical setting. Additionally, a multicenter prospective study approach with several TBD disease patient groups would aid in a health economic assessment and awareness for TBD diagnosis with TICKPLEX^®^. Furthermore, a systematic investigation is required to assess the significance and prevalence of TBD patients with an IgM or IgG-positive immune response to every microbial protein (universally positive).

In conclusion, the present study makes evident that the clinical performance of Borrelia spirochete species and Borrelia persistent forms on TICKPLEX^®^ is in-line with the industry standard PA, NA, and OA. Additionally, the unique Borrelia protein combination in TICKPLEX^®^ can reduce the need from four tests for a LD diagnosis to just one test. Furthermore, in a routine clinical lab, a multiplex and multifunctional test can help detect TBD-related coinfections and opportunistic microbes in LD patients. Moreover, the screening of febrile or summer flu patients for TBDs could be a missed opportunity at reducing misdiagnosed and undiagnosed patient cases.

## Figures and Tables

**Figure 1 tropicalmed-06-00038-f001:**
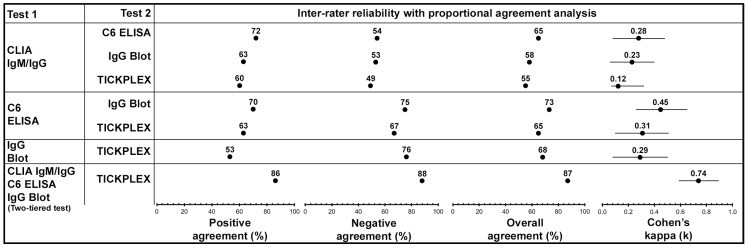
TICKPLEX^®^ can aid replace the need for four Lyme disease diagnostic tests, as the index test clinical performance substantially agrees with the Centers for Disease Control and Prevention (CDC) two-tier system. The collective immunoglobulin M/immunoglobulin G (IgM/IgG) inter-rater reliability (i.e., Cohen’s *k*) and proportional agreement analysis (i.e., positive, negative, and overall agreement) among reference tests and between the reference tests with the index test. Herein, reference tests refer to Diasorin LIAISON® Borrelia chemiluminescence immunoassay (CLIA), Immunetics® C6 Lyme ELISATM (C6 ELISA), and Mikrogen Diagnostik *recom*Bead Borrelia IgG 2.0 (IgG Blot). Similarly, the index test refers to TICKPLEX^®^ PLUS (TICKPLEX^®^). Further, Cohen’s *k* ranges from −1 to +1, wherein *k* values ≤ 0 indicates no agreement, 0.01–0.20 as none to a slight agreement, 0.21–0.40 as fair agreement, 0.41–0.60 as moderate agreement, 0.61–0.80 as substantial agreement, and 0.81–1.00 as almost perfect agreement. The present figure uses the reference and index test results from Lyme disease-positive (*n* = 48) and -negative (*n* = 30) groups.

**Figure 2 tropicalmed-06-00038-f002:**
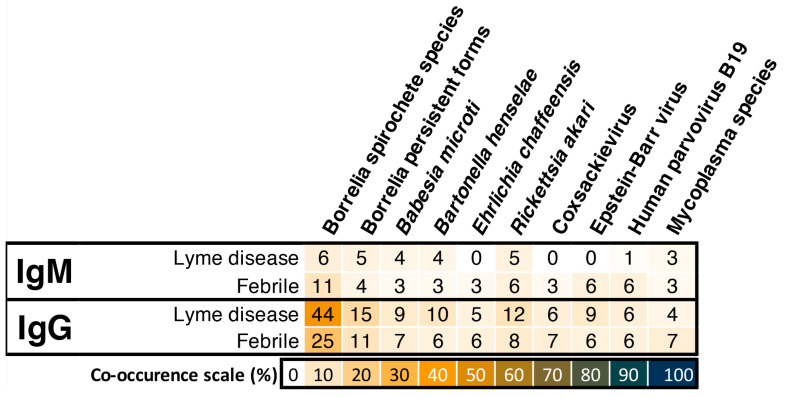
Screening of febrile patients for tick-borne diseases could be a missed opportunity at reducing misdiagnosed and undiagnosed patient cases, as their positive IgM and IgG immune response percentages are similar to the Lyme disease group. Borrelia spirochete species and Borrelia persistent forms refer to *Borrelia burgdorferi sensu stricto*, *Borrelia afzelii*, and *Borrelia garinii* in spirochete and persistent forms, respectively. Similarly, Mycoplasma species refers to *Mycoplasma fermentans* and *Mycoplasma pneumoniae.*

**Figure 3 tropicalmed-06-00038-f003:**
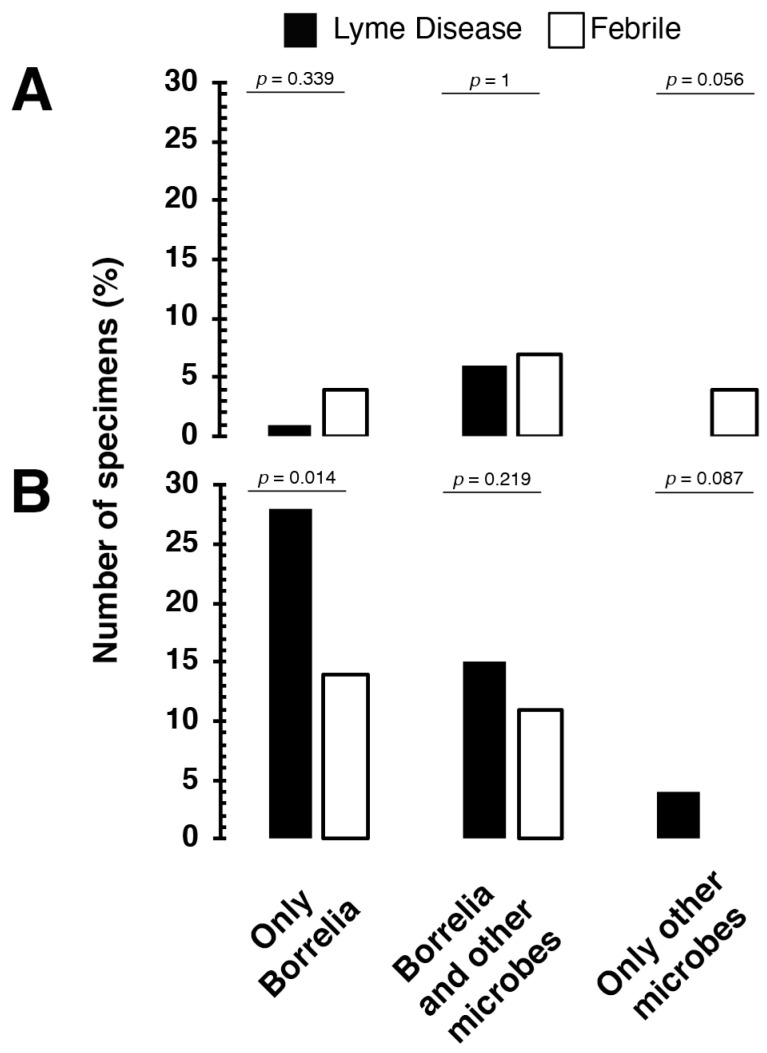
Lyme disease and febrile patients produce (**A**) IgM and (**B**) IgG immune responses to Borrelia and multiple coinfections and opportunistic microbes related to tick-borne diseases. Other microbes refer to *Babesia microti*, *Bartonella henselae*, *Ehrlichia chaffeensis*, *Rickettsia akari*, Coxsackievirus, Epstein–Barr virus, Human parvovirus B19, *Mycoplasma fermentans*, and *Mycoplasma pneumoniae* in the index test. The *p*-value originates from the Fisher’s exact test that was used to assess the statistical differences in IgM or IgG immune responses between Lyme disease (LD) (positive and negative) and febrile patient groups. The two-tailed *p*-values for the Fisher’s exact test were calculated using GraphPad (https://www.graphpad.com/quickcalcs/contingency1/ (accessed on 28 May 2019)). The Fisher’s exact test results with *p*-values < 0.05 were considered statistically associated or dependent.

## Data Availability

Data was contained within the article or Appendix A. All normalized optical density values resulting from the index test were provided in the Appendix A.

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
