# Peer review of "Assessing the Need for Multiplex and Multifunctional Tick-Borne Disease Test in Routine Clinical Laboratory Samples from Lyme Disease and Febrile Patients with a History of a Tick Bite"

_tropicalmed, 2021, doi:10.3390/tropicalmed6010038_

Round 1

Reviewer 1 Report

Estimated Authors,

Estimated Editors,

I've read with great interest this paper on the need for multiplex and multifunctional tick-borne disease test in routine clinical laboratory samples from Lyme disease and summer flu patients. The study group lead by Garg has reported their preliminary report on the use of a multiple diagnostic test, and their results hints towards a future approach to tick borne diseases including a larger array of testing. As tick may be vectors for several pathogens, this may significantly improve our approach to suspected cases.

However, in my opinion some improvements are required.

Honestly, I had some difficulties approaching this paper, as aims/methods were somewhat confused by the sequence of the "methods" section. I would suggest to move section 2.4 at the beginning of the methods section, in order to focus the attention of the readers on the instruments you would employ and how you assessed the diagnosis of Tick Borne Diseases.

Moreover, I would suggest to include some further insights on the characteristics of the subjects the specimens were taken from. Some details are provided at the beginning of section 3, but it should expanded (if possible).

Eventually (some very minor improvements)

  • please explain whether it was a convenience sampling (i.e. no more specimens were available) or did you perform a preventive power analysis (in case, report it)
  • please explain what the "summer flu" you assessed actually was. I mean: I understood SF as an influenza like illness (i.e. a syndrome) occurring during the summer season, but SF may be also an alternative term for syndrome associated with Lyme disease but without a confirmative laboratory analysis. In case I was right, please either reframe "summer flu" as "summer influenza" or explain your working definition of SF.

After such improvements, I think that your paper may be eventually accepted.

Author Response

Point-by-point response for reviewer #1

Reviewer’s point #1

Honestly, I had some difficulties approaching this paper, as aims/methods were somewhat confused by the sequence of the "methods" section. I would suggest to move section 2.4 at the beginning of the methods section, in order to focus the attention of the readers on the instruments you would employ and how you assessed the diagnosis of Tick Borne Diseases.

Author response #1         

Section 2.4 has been moved to the beginning of the methods section as section “2.1. Index test and interpretation.”

Reviewer’s point #2

Moreover, I would suggest to include some further insights on the characteristics of the subjects the specimens were taken from. Some details are provided at the beginning of section 3, but it should expanded (if possible).

Author response #2

All available information concerning the specimens have been included sections 2 and 3. Information available on the leftover and deidentified samples were limited. Additionally, we recognize the need for a prospective study in the discussion section as an improvement in study design for future studies.

Reviewer’s point #3

Please explain whether it was a convenience sampling (i.e. no more specimens were available) or did you perform a preventive power analysis (in case, report it).

Author response #3

We have clarified at the beginning of section 3 that the specimens were collected amid routine clinical diagnostic services (Convenience sampling) at United Medix Laboratory.

Reviewer’s point #4

Please explain what the "summer flu" you assessed actually was. I mean: I understood SF as an influenza like illness (i.e. a syndrome) occurring during the summer season, but SF may be also an alternative term for syndrome associated with Lyme disease but without a confirmative laboratory analysis. In case I was right, please either reframe "summer flu" as "summer influenza" or explain your working definition of SF.

Author response #4

The summer flu patient group has been reframed as a febrile patient group to better represent the specimens. Section 2.4 in the revised manuscript provides a working definition for the febrile patient group. As a consequence, Figures 2, 3, S1, S2, and Tables S1 and S2 have been updated.

Reviewer 2 Report

Manuscript untitled „Assesing the need for multiplex and multifunctional tick-borne disease test in routine clinical laboratory samples from Lyme Disease and summer flu patientspoint out on very important problem with making diagnosis for patients after tick bite. Present results trying to show utility of TICKPLEX PLUS for Lyme Disease diagnostic instead of four tests. According to authors additional advantage of TICKPLEX PLUS is to detected simultaneously immune response for other tick-borne diseases.

Article is divided into clear parts and well organized. While number of patients is only for predominately study. This results should be performed on more patients than in group of 150 people, and only 48 with LD. Beside patients were involved to LD group even if they have only one from four diagnostic test positive to Borrelia antibodies. Even if we are aware that those tests used different proteins is too small number. This defect is marked by authors in discussion section and in conclusions.

Figrure 1 shows statistical association between Lyme disease and summer flu in IgM in EBV virus and in IgG in Borrelia infection what was previously well known. There were no differences in both group of patients and IgM or/and IgG response for Babesia microti, Bartonella henselae, Ehrlichia chaffensis, Rickettsia akari, Coxsackievirus, Human parvovirus B19 and Mycoplasma species.

Statistical analysis is well prepared but should involved large number of patients. Results are presented on many graphs, that is why they are well presented.

Discussion is carefully performed. Authors in detailed way describe advantages and disadvantages of TICKPLEX test and obtained results.

It is one of manuscript which indicates the need of co-infection detection after tick bite. As long as we will diagnose only one or two tick-borne diseases in endemic regions we will misdiagnosed large group of patients.

To sum up, I give a positive opinion about manuscript untitled „Assesing the need for multiplex and multifunctional tick-borne disease test in routine clinical laboratory samples from Lyme Disease and summer flu patients”, but I strongly recommended performed analysis in larger group of patients.

Author Response

Point-by-point response for reviewer #2

Reviewer’s point #1

To sum up, I give a positive opinion about manuscript untitled „Assesing the need for multiplex and multifunctional tick-borne disease test in routine clinical laboratory samples from Lyme Disease and summer flu patients”, but I strongly recommended performed analysis in larger group of patients.

Author response #1

We agree with the reviewer’s assessment concerning the sample size in our study. Our discussion section recognizes the need for an overall increase in the sample size and a multi-centre prospective study design for future research.